# Biologically-Constrained Multi-Label Classification with Learnable Domain Knowledge

**Nabil Mouadden**[1]                                        NABIL.MOUADDEN@CENTRALESUPELEC.FR
**Veronique Verge**[2]                                        VERONIQUE.VERGE@GUSTAVEROUSSY.FR
**Ahmadreza Arbab**[2]                                        AHMADREZA.ARBAB@GUSTAVEROUSSY.FR
**Jean-Baptiste Micol**[2]                                  JEANBAPTISTE.MICOL@GUSTAVEROUSSY.FR
**Elsa Bernard**[2]                                              ELSA.BERNARD@GUSTAVEROUSSY.FR
**Aline Renneville**[2]                                      ALINE.RENNEVILLE@GUSTAVEROUSSY.FR
**Stergios Christodoulidis**[1]            STERGIOS.CHRISTODOULIDIS@CENTRALESUPELEC.FR
**Maria Vakalopoulou**[1]                      MARIA.VAKALOPOULOU@CENTRALESUPELEC.FR

[1] *MICS, CentraleSupelec, Paris-Saclay University, France*
[2] *Gustave Roussy, Villejuif, France*

**Editors:** Under Review for MIDL 2025

## Abstract

Although recent foundation models trained in a self-supervised setting have shown promise in cellular image analysis, they often produce biologically impossible predictions when handling multiple concurrent abnormalities. This is a problem, as the biological information that may be needed for the different clinical-oriented problems is not directly presented in the images. In this study, we present a novel and modular approach to enforce biological constraints in multi-label medical imaging classification. Building on the powerful and rich representations of the DinoBloom hematological foundation model, our method combines learnable constraint matrices with adaptive thresholding, effectively preventing contradictory predictions while maintaining high sensitivity. Extensive experiments on three datasets, two public and one in-house on neutrophil classification, demonstrate significant improvements over different foundation models and the state-of-the-art methods. Through detailed ablation studies and hyperparameter interpretation, we show that our approach successfully captures biological relationships between different abnormalities.

## 1. Introduction

Recent advances in foundation models have revolutionized medical image analysis, with DinoBloom emerging as a powerful foundation model specifically designed for hematological image analysis (10). However, while DinoBloom excels at general feature extraction from blood cell images, integrating explicit biological constraints and handling multiple concurrent abnormalities remains an open challenge. This limitation is particularly evident in clinical settings where predictions must adhere to known biological impossibilities and relationships.

Neutrophils are a type of white blood cells, and they form the most abundant type of granulocytes of all white blood cells in humans. Neutrophil morphology analysis plays a crucial role in hematological diagnosis, serving as a fundamental tool for identifying various blood disorders and infections (12). Traditional manual classification of neutrophil abnormalities is time-consuming and subject to significant inter-observer variability, with reported concordance rates as low as 60% among experts (13). While recently deep learning approaches have shown remarkable promise in

medical image analysis (14), existing methods fail to address two critical challenges unique to neutrophil classification: (1) the complex interdependencies between multiple concurrent abnormalities, and (2) the need to enforce explicit biological constraints in predictions (15).

In this study, we propose a novel, differentiable framework that enhances foundation models with learnable biological constraints. The main contributions of this study are two folds: *(i)* we introduce a learnable constraint satisfaction module that automatically discovers and enforces biological relationships while maintaining end-to-end differentiability, *(ii)* we propose an adaptive thresholding mechanism that dynamically adjusts to varying degrees of abnormality manifestation. Our extensive experiments and ablations using the DinoBloom foundation model for neutrophil morphology analysis, which included three different datasets with varying numbers of abnormalities, highlight the superiority of our method with respect to the state of the art.

## 2. Related Work

**Foundation Models in Hematology.** Hematological image analysis has recently embraced large-scale self-supervision and transformers. Early efforts, such as Matek et al. (15) employed supervised convolutional networks on smaller datasets. More recent approaches leverage large data corpora and attention mechanisms: DinoBloom (11) emerged as a specialized foundation model for white blood cell (WBC) morphology, while Wang et al. (3) proposed unsupervised contrastive transformers for histopathological images. The introduction of domain-specific transformer architectures by Filiot et al. (2) further advanced the field through masked image modeling. Although these approaches learn strong representations, they do not explicitly incorporate domain-specific constraints or handle biologically impossible co-occurrences.

**Multi-Label Classification in Medical Imaging.** Many medical tasks inherently involve multi-label outputs, as conditions often co-exist or overlap. Traditional solutions like binary relevance overlook label correlations, which has prompted research into more holistic methods. Wang et al. (3) proposed unsupervised contrastive transformers for histopathological images, while Wang et al. (19) proposed clustering-guided contrastive learning for cell images. Chen et al. (8) explored attention mechanisms, yet such methods often assume static or binary inter-label relationships, and do not address clinical uncertainty where co-occurrence can be probabilistic.

**Domain-Constrained Learning.** Incorporating domain knowledge into deep models has gained traction in medical imaging (6; 5), showing that enforcing biologically meaningful constraints can improve both performance and interpretability. However, many techniques rely on rigid rules or postprocessing steps.(15) have shown the importance of incorporating morphological constraints in leukemia cell classification. Beyond hematology, these studies have demonstrated the value of domain constraints across various medical imaging applications, though they often lack the flexibility to capture probabilistic relationships that arise in real-world clinical settings. In contrast, our approach unifies a powerful transformer-based hematology backbone with learnable constraint matrices and adaptive thresholding, thus capturing both deterministic incompatibilities and nuanced, uncertain relationships that arise in real-world clinical settings.

## 3. Methods

### 3.1. Problem Formulation

Given an input image $x \in \mathbb{R}^{H \times W \times 3}$, where $H$ and $W$ represent the height and width of the image respectively, we aim to predict a set of binary labels $\mathbf{y} = [y_1, ..., y_K] \in \{0, 1\}^K$, where $K$ is the number of possible classes. The prediction must satisfy two types of biological constraints: *(i)* *mutual exclusivity constraints:* defined as $\mathcal{C}_{mu} = \{(i, j)|y_i \cdot y_j = 0\}$, where $(i, j)$ represents pairs

of classes that cannot co-exist, and *(ii) co-occurrence constraints:* defined as $\mathcal{C}_{co} = \{(i, j, c)|y_i = 1 \implies P(y_j) \geq c\}$, where $c \in [0, 1]$ is a threshold probability and models the increased or decreased likelihood of the presence of one class with respect to the rest.

Traditional approaches perform classification in isolation, without considering the rich domain knowledge that biologists leverage in their decision-making process (24; 7). However, in practice, experts rely on their deep understanding of biological relationships and manifestations - they know that certain abnormalities often co-occur, that some conditions are mutually exclusive, and that the same abnormality can present with varying degrees of severity (13; 12). Moreover, biologists can discover new relationships between conditions through observations and adjust their confidence based on the strength of different markers (4; 6). Our method aims to emulate this expert reasoning by incorporating learnable biological constraints and adaptive decision thresholds. Our method addresses these limitations by integrating the $\mathcal{C} = \mathcal{C}_{mu} + \mathcal{C}_{co}$ into the training process through learnable constraint and uncertainty-aware adaptive thresholding. An overview of the entire method is presented in Figure 1.

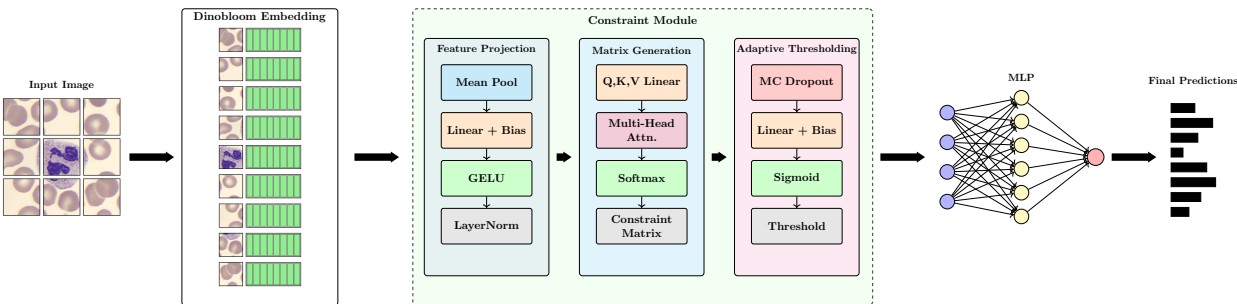

Figure 1: **Biologically-Constrained Multi-Label Classification Architecture.** The proposed model consists of three main components within the constraint module: (A) Feature projection with pooling and normalization, (B) Constraint matrix generation using attention mechanism, and (C) Adaptive thresholding with Monte Carlo sampling.

## 3.2. Learnable Constraint Module

We introduce a learnable constraint module that enhances the model's ability to capture and enforce biological relationships. This module consists of three key components designed to work together:

**Projection Layer.** The feature projection layer transforms high-dimensional features into a space more suitable for learning biological constraints. We start with the features $\mathbf{h}_L \in \mathbb{R}^{N \times d}$, where $N$ is the number of image patches (196 for 224×224 images with 16×16 patches) and $d$ is the feature dimension. These features are processed through a projection layer to obtain $\mathbf{f} \in \mathbb{R}^d$, which represents a condensed feature vector: $\mathbf{f} = \text{LayerNorm}(\text{GELU}(\mathbf{W}_p\text{Pool}(\mathbf{h}_L) + \mathbf{b}_p))$. where Pool$(\cdot)$ performs mean pooling over the N patches to create a single d-dimensional feature vector, $\mathbf{W}_p \in \mathbb{R}^{d \times d}$ is a learnable weight matrix that projects the pooled features while preserving dimensionality and $\mathbf{b}_p \in \mathbb{R}^d$ is a learnable bias vector, GELU (Gaussian Error Linear Unit) is a smooth activation function that helps maintain gradient flow and LayerNorm normalizes the features across the feature dimension, stabilizing training by ensuring consistent feature scales. In practice, $\mathbf{f}$ is a $d$-dimensional feature vector that aggregates the patch-level information from DinoBloom into a single vector representing the entire cell, and it projects the features into a space where biological relationships can be more easily learned through the constraint mechanism.

**Constraint Matrix Generation.** For the constraint matrix, we employ a transformer-based multi-head attention mechanism to learn the relationships between different abnormalities. This

mechanism allows the model to learn both positive (co-occurrence) and negative (mutual exclusivity) relationships dynamically: $\mathbf{Q} = \mathbf{W}_q\mathbf{f}, \quad \mathbf{K} = \mathbf{W}_k\mathbf{f}, \quad \mathbf{V} = \mathbf{W}_v\mathbf{f}$, where $\mathbf{W}_q, \mathbf{W}_k, \mathbf{W}_v \in \mathbb{R}^{K \times d}$ are learnable parameter matrices that transform the features into query, key, and value representations and $K$ is the number of possible abnormalities. The constraint matrix $\mathbf{R} \in \mathbb{R}^{K \times K}$ captures pairwise relationships between abnormalities and it is then computed using scaled dot-product, $\mathbf{R} = \text{softmax}\left(\frac{\mathbf{Q}\mathbf{K}^T}{\sqrt{d}}\right)\mathbf{V}$.

**Adaptive Thresholding.** We introduce an uncertainty-aware adaptive thresholding mechanism that adjusts decision boundaries based on prediction confidence and Monte Carlo dropout sampling. During inference, we apply dropout (with rate 0.5) to the feature vector $f$ and perform $M$ stochastic forward passes through the classification layer: $\hat{\mathbf{y}}^{(m)} = \sigma(\mathbf{W}_c\text{Dropout}(\mathbf{f}) + \mathbf{b}_c)$ where $m = 1, \ldots, M$ with $M = 50$ Monte Carlo samples. The predictive uncertainty for each class is computed as: $U_i = \frac{1}{M}\sum_{m=1}^{M}(\hat{y}_i^{(m)} - \bar{y}_i)^2$, where $\bar{y}_i = \frac{1}{M}\sum_{m=1}^{M}\hat{y}_i^{(m)}$ is the mean prediction for class $i$. The adaptive threshold for each class is then calculated as: $t_i = \alpha_i \cdot t_{\text{base}} + \beta_i \cdot U_i + \delta_i \cdot p_i$, where each term serves a specific purpose: $t_{\text{base}}$ provides a starting point that can be adjusted up or down, $U_i$ incorporates model uncertainty to require higher confidence thresholds when predictions are uncertain, and $p_i$ accounts for class frequency in the training data to adjust for class imbalance. The learnable parameters $\alpha_i, \beta_i, \delta_i$ allow the model to fine-tune the importance of each component for different abnormalities - crucial since some morphological features require more certainty for positive prediction than others (e.g., subtle chromatin changes versus obvious hypersegmentation).

## 3.3. Training Strategy

We employ a single training phase where the backbone is frozen while other components are trained end-to-end with distinct learning rates. The total loss function combines multiple terms:

$$\mathcal{L}_{\text{total}} = \mathcal{L}_{\text{BCE}} + \lambda_1\mathcal{L}_{\text{con}} + \lambda_2\mathcal{L}_{\text{unc}} + \lambda_3\mathcal{L}_{\text{entropy}} \tag{1}$$

where:

$$\mathcal{L}_{\text{BCE}} = -\frac{1}{N}\sum_{i=1}^{N}\sum_{k=1}^{K}[y_{ik}\log(\hat{y}_{ik}) + (1 - y_{ik})\log(1 - \hat{y}_{ik})] \tag{2}$$

$$\mathcal{L}_{\text{con}} = \|\mathbf{R}\mathbf{R}^T - \mathbf{C}\|_F^2 + \alpha\|\mathbf{R}\|_1 \tag{3}$$

$$\mathcal{L}_{\text{unc}} = \frac{1}{NK}\sum_{i=1}^{N}\sum_{k=1}^{K}[\text{KL}(p_{ik}\|\hat{p}_{ik}) + \beta\max(0, U_{ik} - \tau)] \tag{4}$$

$$\mathcal{L}_{\text{entropy}} = -\frac{1}{K^2}\sum_{i=1}^{K}\sum_{j=1}^{K}[\mathbf{R}_{ij}\log(\mathbf{R}_{ij})] \tag{5}$$

Here, $\mathbf{C} \in \mathbb{R}^{K \times K}$ is the prior constraint matrix encoding known biological relationships where $C_{ij} = -1$ for mutually exclusive pairs (e.g., hypersegmentation vs hyposegmentation), $c_{ij}$ for co-occurring abnormalities with $c_{ij}$ being their co-occurrence strength, and $c_{ij} = 0$ for unrelated pairs. The product $\mathbf{R}\mathbf{R}^T$ captures both direct and indirect (transitive) relationships between abnormalities through matrix multiplication, which when compared to $\mathbf{C}$ using the Frobenius norm ensures these learned relationships match known biological constraints. The L1 norm $\|\mathbf{R}\|_1$ encourages the model to learn sparse relationships by pushing non-essential elements of $\mathbf{R}$ towards zero. Given the

inherent uncertainty in biological relationships, we deliberately set small values for the hyperparameters $\lambda_1, \lambda_2, \lambda_3$ to prevent the model from enforcing strong associations when there is no clear biological evidence. This conservative approach is particularly important as relationships between morphological features can be context-dependent and vary across different pathological conditions. Similarly, $\alpha$ and $\beta$ are set to small values to maintain flexibility in the learned relationships while still providing enough regularization to prevent spurious correlations.

### 3.4. Implementation Details

For this study and for efficiency, we utilize DinoBloom-S (10) as our feature extraction backbone, which consists of a Vision Transformer (ViT) architecture pretrained on a large corpus of hematological images. The $\mathbf{h}_L$ for the DinoBloom-S model has an embedding dimension of $d = 384$. For more details, please check the original DinoBloom paper. Moreover, the uncertainty threshold $\tau$ was set to 0.2 in our experiments, the base threshold $t_{\text{base}}$ was set to 0.5, the loss weights as $\lambda_1 = 0.1$, $\lambda_2 = 0.1$, $\lambda_3 = 0.01$ and the hyperparameters $\alpha = 0.01$, $\beta = 0.1$ respectively. During training, different learning rates are employed for each component while keeping the DinoBloom feature extractor frozen: the constraint module uses $\eta_c = 1e - 4$, the uncertainty estimation components use $\eta_u = 5e - 5$, and the classification head uses $\eta_h = 1e - 4$. For the training, we used an AdamW optimizer, and the training was performed on 1 NVIDIA A100 GPUs, taking approximately 30 minutes for convergence. Finally, all the information about the constraint matrices ($C$) per dataset is presented in Appendix A.

## 4. Experimental Results

### 4.1. Datasets

We evaluate our method on three datasets, two public and one in-house. **AML Matek Dataset.** (15) consists of 18,365 expert-labeled single-cell images with 15 morphological classes and multiple concurrent abnormalities. Following the original split, we use 15,827 images from 100 AML and 100 non-AML patients for training, with the remaining 2,538 images from 40 patients held out for testing. **BMC Dataset.** (16) contains 171,373 cells from bone marrow smears of 945 patients, annotated with 21 distinct cell types. The dataset is divided following the original paper's protocol: 137,098 cells (756 patients) for training and 34,275 cells (189 patients) for testing. This split ensures patient-level separation between train and test sets. The dataset is highly imbalanced, with some rare cell types having as few as 8 samples. **GR-Neutro Dataset.** Our dataset comprises 1,934 high-resolution microscopy images of neutrophils, including both normal cells (878 images) and various abnormalities (582 images) without any patient level information. The dataset was split into training (1,455 images) and test (479 images) sets using stratified sampling to maintain class distribution. More details about the dataset and its classes are presented in Appendix B.

All datasets were preprocessed following the same protocol: images were resized to 224×224 pixels and normalized using mean and standard deviation computed from the training set. To handle class imbalance, we employed a combination of techniques including oversampling of minority classes using SMOTE (Synthetic Minority Over-sampling Technique) (9), undersampling of majority classes using random undersampling, and class weights in the loss function proportional to the inverse of class frequencies. For DinoBloom comparison, we used their recommended preprocessing pipeline. Standard augmentations including random horizontal flips, rotations, and color jittering were applied during training, with more aggressive augmentation strategies applied to underrepresented classes to further address imbalance.

| Method | Normal | | Chromatin | | Dohle | | Hypergr. | | Hyperseg. | | Hypogr. | | Hyposeg. | | Overall | |
|---|---|---|---|---|---|---|---|---|---|---|---|---|---|---|---|---|
| | wF1 | bAcc | wF1 | bAcc | wF1 | bAcc | wF1 | bAcc | wF1 | bAcc | wF1 | bAcc | wF1 | bAcc | wF1 | bAcc |
| DINOv2 ViT-S/14 | 0.72 | 0.70 | 0.43 | 0.41 | 0.76 | 0.74 | 0.94 | 0.92 | 0.79 | 0.77 | 0.80 | 0.78 | 0.84 | 0.82 | 0.68 | 0.66 |
| DINOv2 ViT-B/14 | 0.71 | 0.69 | 0.48 | 0.46 | 0.77 | 0.75 | 0.92 | 0.90 | 0.81 | 0.79 | 0.82 | 0.80 | 0.83 | 0.81 | 0.70 | 0.68 |
| DINOv2 ViT-L/14 | 0.72 | 0.70 | 0.49 | 0.47 | 0.77 | 0.75 | 0.89 | 0.87 | 0.83 | 0.81 | 0.81 | 0.79 | 0.84 | 0.82 | 0.71 | 0.69 |
| DinoBloom-S | 0.86 | 0.84 | 0.55 | 0.53 | 0.86 | 0.84 | 0.94 | 0.92 | 0.89 | 0.87 | 0.90 | 0.88 | 0.84 | 0.82 | 0.85 | 0.83 |
| DinoBloom-B | 0.87 | 0.85 | 0.61 | 0.59 | 0.87 | 0.85 | 0.92 | 0.90 | 0.91 | 0.89 | 0.91 | 0.89 | 0.83 | 0.81 | 0.86 | 0.84 |
| DinoBloom-L | 0.88 | 0.86 | 0.63 | 0.61 | 0.87 | 0.85 | 0.91 | 0.89 | 0.91 | 0.89 | 0.90 | 0.88 | 0.84 | 0.82 | 0.86 | 0.84 |
| CTransPath | 0.80 | 0.78 | 0.52 | 0.50 | 0.83 | 0.81 | 0.88 | 0.86 | 0.80 | 0.78 | 0.82 | 0.80 | 0.83 | 0.81 | 0.74 | 0.72 |
| Phikon ViT-B | 0.83 | 0.81 | 0.54 | 0.52 | 0.85 | 0.83 | 0.88 | 0.86 | 0.82 | 0.80 | 0.85 | 0.83 | 0.83 | 0.81 | 0.76 | 0.74 |
| **Ours** | **0.99** | **0.97** | **0.87** | **0.85** | **0.95** | **0.93** | **1.00** | **1.00** | **0.97** | **0.95** | **0.90** | **0.88** | **0.90** | **0.88** | **0.94** | **0.92** |

Table 1: Performance comparison on the GR-Neutro dataset showing both weighted F1-score (wF1) and balanced accuracy (bAcc) for each cell type. Best results are shown in **bold**, second best are underlined.

| Dataset | Metric | DINOv2 ViT-S/14 | DINOv2 ViT-B/14 | DINOv2 ViT-L/14 | CTransPath | Phikon ViT-B | DinoBloom-L | Ours |
|---|---|---|---|---|---|---|---|---|
| AML Matek | wF1 | 0.88 | 0.88 | 0.89 | 0.88 | 0.88 | 0.91 | **0.95** |
| | bAcc | 0.82 | 0.82 | 0.84 | 0.83 | 0.83 | 0.86 | **0.91** |
| BMC | wF1 | 0.68 | 0.71 | 0.71 | 0.74 | 0.73 | 0.85 | **0.89** |
| | bAcc | 0.45 | 0.49 | 0.48 | 0.52 | 0.54 | 0.64 | **0.75** |
| **Method** | | Zunair et al.[17] | Chang et al.[18] | García-García et al. [20] | Kassahun et al.[21] | Wang et al. [22] | Li et al.[23] | **Ours** |
| AML Matek | wF1 | 0.93 | 0.94 | 0.94[a] | — | — | — | **0.95** |
| | bAcc | — | — | — | — | — | — | **0.91** |
| BMC | wF1 | — | — | — | 0.85 | 0.87 | 0.86 | **0.89** |
| | bAcc | — | — | — | 0.73 | 0.74 | 0.73 | **0.75** |

Table 2: Performance on the AML Matek ([15]) and BMC ([16]) datasets. Comparisons with different methods and models are provided. [a] This refers to *blast vs. non-blast* classification, not the full 15-class AML Matek scheme.

## 4.2. Comparison with other models

We compare our approach with several state-of-the-art methods, including DINOv2 ([1]) models (ViT-S/14, ViT-B/14, ViT-L/14) pretrained on general image data, DinoBloom ([10]) models (S, B, L) specifically trained on hematological data, CTransPath ([3]) trained on pathology image data and the Phikon ViT-B ([2]) trained on histopathology images. For the comparison of these models we used weighted F1-score (wF1) and balanced accuracy (bAcc). We summarize the results in Table 1 for the GR-Neutro and Table 2 for the public dataset. Our proposed method consistently outperforms all the baselines compared in all datasets. Please note that our method can be adapted to any backbone presented before, making it very modular and easy to use.

Moreover, Table 2 summarize the comparison of our approach against several state-of-the-art methods on AML Matek ([15]) and BMC ([16]), including recent Transformer-based techniques ([17]; [18]; [20]), multi-task and self-supervised methods ([21]; [22]; [23]). Our approach on AML Matek achieves a wF1-score of 0.95 and a bAcc of 0.91, outperforming prior methods such as Chang et al. ([18]) and Garc'ıa-Garc'ıa et al. ([20]), both of which reported wF1 up to 0.94. Notably, those references do not report balanced accuracy, making our 0.91 bAcc a strong indicator of performance on less frequent cell types. In addition, for the BMC we obtain with our method a wF1 of 0.89 with a 0.75 of bAcc. This outperforms multi-task and self-supervised approaches like Kassahun et al. ([21]) (0.85 wF1, 0.73 bAcc) and Wang et al. ([22]) (0.87 wF1, 0.74 bAcc), confirming that our constraint-based approach manages the extensive class imbalance of the BMC dataset effectively.

The per-class performance analysis on the GR-Neutro dataset presented in Table 1 and Appendix C reveals the robustness of our approach across different abnormality types. Most notably,

our method achieves perfect accuracy for hypergranulation detection and near-perfect performance for normal cell classification. The substantial improvements in challenging cases like chromatin condensation (0.87 vs 0.63 in DinoBloom-L) and Dohle bodies (0.95 vs 0.87) demonstrate the effectiveness of our constraint-based learning approach in handling subtle morphological variations. The consistent performance across all classes, including traditionally challenging ones like hyposegmentation and hypogranulation, highlights the balanced nature of our approach. Some examples from the attention maps obtained by the constraint attention are presented in Appendix D.

### 4.3. Ablation Study

| Model Variant | Accuracy | F1 | AUC | Spec. | Sens. |
|---|---|---|---|---|---|
| Base Model | 0.81 | 0.79 | 0.83 | 0.82 | 0.80 |
| + Learnable constraints | 0.88 | 0.87 | 0.90 | 0.89 | 0.88 |
| + Adaptive Thresholds | 0.94 | 0.93 | 0.95 | 0.93 | 0.94 |
| **Constraint Layer Variants** | | | | | |
| Fixed Rules | 0.85 | 0.83 | 0.86 | 0.84 | 0.85 |
| Binary Constraints | 0.86 | 0.84 | 0.87 | 0.85 | 0.86 |
| Soft Constraints (Ours) | **0.88** | **0.87** | **0.90** | **0.89** | **0.88** |
| **Threshold Mechanism** | | | | | |
| Static Threshold | 0.89 | 0.88 | 0.90 | 0.88 | 0.89 |
| Class-Specific | 0.90 | 0.89 | 0.91 | 0.90 | 0.90 |
| Uncertainty-Aware (Ours) | **0.94** | **0.93** | **0.95** | **0.93** | **0.94** |

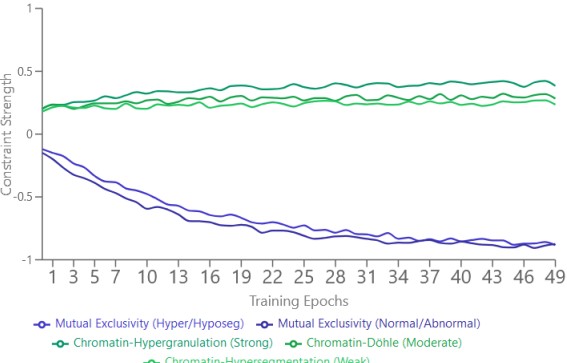

Figure 2: **Left:** Ablation study analysis showing the impact of each model component. **Right:** Evolution of learned constraint relationships during training, showing how the model discovers biologically meaningful patterns - mutual exclusivity constraints converge to strong negative values, while co-occurrence relationships stabilize at different positive strengths based on their biological significance.

To highlight the effectiveness of each component of our method, we conducted an extensive ablation study summarized in Figure 2 (Left). The results are presented in three distinct evaluation settings to demonstrate both the cumulative and individual impacts of our key components. First, we evaluate the progressive addition of components, starting with a base model (0.81 accuracy), then adding learnable constraints (improving to 0.88), and finally incorporating adaptive thresholds (reaching 0.94). This demonstrates the cumulative benefit of our complete architecture. For constraint mechanisms, we compare three variants in isolation: (i) *fixed rules*, which uses manually predefined constraints (e.g., hardcoding that hyper/hyposegmentation cannot co-occur), achieving 0.85 accuracy, (ii) *binary constraints*, where relationships between abnormalities are limited to strict 0/1 values, reaching 0.86 accuracy, and (iii) our *soft constraints* approach, which learns continuous values between 0 and 1 to represent relationship strengths, achieving 0.88 accuracy. Similarly, for thresholding mechanisms, we evaluate three approaches: (i) *static threshold*, using a fixed threshold (0.5) for all classes, achieving 0.89 accuracy, (ii) *class-specific threshold*, where each class has its own learned threshold, improving to 0.90 accuracy, and (iii) our *uncertainty-aware* approach, which dynamically adjusts thresholds based on prediction confidence, reaching 0.94 accuracy when integrated with the full model. These results demonstrate that both components contribute significantly to model performance, with each achieving their best results when combined in the full architecture.

The evolution of constraints during training (Figure 2, Right) demonstrates how our model discovers and enforces biological relationships in the GR-Neutro Dataset. The mutual exclusivity constraints between hypersegmentation and hyposegmentation converge to strong negative

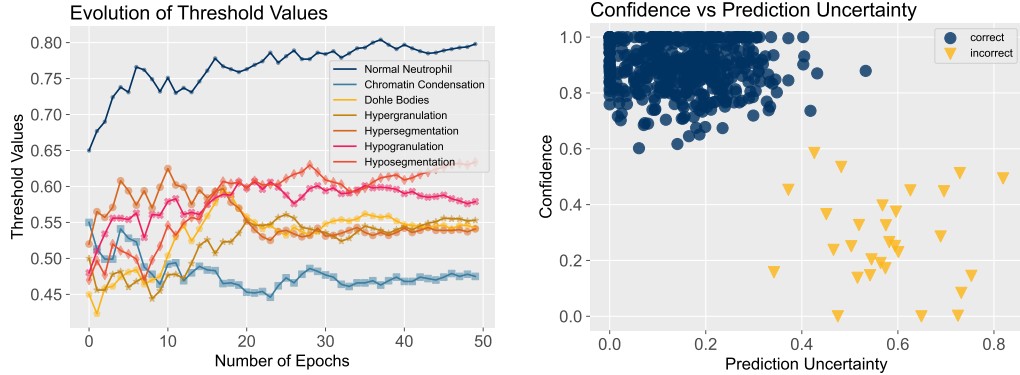

Figure 3: **Left:** Evolution of class-specific adaptive thresholds during training, showing how thresholds adapt to different neutrophil abnormalities based on their characteristics for the GR-Neutro, and **Right:** Relationship between model uncertainty and prediction confidence, showing clear separation between correct (blue) and incorrect (yellow) predictions for the GR-Neutro Dataset.

values $\leq$-0.8, indicating the model's clear understanding of incompatible cell states. Moreover, (Figure 2, Right), co-occurrence constraints show more nuanced behavior ($\geq 0.45$ for chromatin-hypergranulation, $\geq 0.35$ for chromatin-Döhle, and $\geq 0.25$ for chromatin-hypersegmentation), reflecting the varying strengths of these biological associations. This hierarchy of learned constraints aligns with clinical observations, where mutual exclusivity represents fundamental biological impossibilities while co-occurrences represent more flexible, probabilistic relationships.

Figure(3, Left) shows the evolution of class-specific adaptive thresholds during training. The thresholds start from a common base value (0.5) and gradually diverge based on class-specific characteristics. Normal neutrophils converge to higher threshold values (around 0.7), reflecting the need for higher confidence when classifying normal cells. In contrast, abnormality thresholds settle at different levels (between 0.4-0.6), with chromatin condensation requiring the lowest threshold (0.4) due to its subtle nature, and more obvious features like hypersegmentation maintaining moderate thresholds (0.55). This adaptive behavior enables the model to account for varying degrees of morphological distinctiveness across different abnormalities while maintaining high specificity. Finally, Figure(3, Right) shows the relationship between prediction confidence and uncertainty, where correct predictions (shown in blue) are clearly separated from incorrect ones (shown in yellow). The model demonstrates high confidence ($\geq$0.8) and low uncertainty ($\leq$0.2) for correct predictions, while incorrect predictions show higher uncertainty ($\geq$0.4) and lower confidence values ($\leq$0.6), validating the effectiveness of our uncertainty estimation approach.

## 5. Conclusion

We presented a novel approach for learning and enforcing biological constraints in multi-label classification of neutrophil abnormalities. Our method not only improves classification accuracy but also provides valuable insights into morphological relationships through learned constraints. The adaptive thresholding mechanism effectively handles varying degrees of abnormality manifestation, while the learnable constraint satisfaction layer prevents biologically impossible predictions. Future work will focus on expanding to larger, multi-center datasets and incorporating temporal dynamics in neutrophil analysis.

## Acknowledgment

We thank Qube Research & Technologies for their support. This work was performed using HPC resources from the "Mesocentre" computing center of CentraleSupelec. This work was partially supported by the ANR Hagnodice ANR-21-CE45-0007 and DAFNI ANR-23-CE45-0029.

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

## Appendix A. Constraint Prior Matrices for Each Dataset

In this appendix, we provide details on the prior constraint matrices $\mathbf{C}$ used for each dataset (AML Matek, BMC, and GR-Neutro). Recall that $C_{ij} = -1$ indicates mutual exclusivity between classes $i$ and $j$, $C_{ij} = 0$ indicates no direct relationship, and $C_{ij} = c_{ij} > 0$ indicates a soft co-occurrence with strength $c_{ij} \in [0, 1]$.

### A.1 GR-Neutro Dataset ($7 \times 7$)

Table 3: Constraint matrix **C** for the GR-Neutro dataset.

|  | Normal | Chromatin | Döhle | Hypergran. | Hyperseg. | Hypogran. | Hyposeg. |
|---|---|---|---|---|---|---|---|
| Normal | 0 | -1 | -1 | -1 | -1 | -1 | -1 |
| Chromatin | -1 | 0 | 0.3 | 0.4 | 0.3 | 0.2 | 0.3 |
| Döhle | -1 | 0.3 | 0 | 0.3 | 0.2 | 0.2 | 0.2 |
| Hypergran. | -1 | 0.4 | 0.3 | 0 | 0.3 | -1 | 0.2 |
| Hyperseg. | -1 | 0.3 | 0.2 | 0.3 | 0 | 0.2 | -1 |
| Hypogran. | -1 | 0.2 | 0.2 | -1 | 0.2 | 0 | 0.2 |
| Hyposeg. | -1 | 0.3 | 0.2 | 0.2 | -1 | 0.2 | 0 |

## A.2 AML Matek Dataset ($15 \times 15$)

Table 4: Constraint matrix **C** for the AML Matek dataset.

|  | Myelo. | Promyelo. | Myelo. | Meta. | Band | Segm. | Eos. | Baso. | Mono. | Lymph. | Plasma | Erythro. | RBC/Plt | Rare | Art. |
|---|---|---|---|---|---|---|---|---|---|---|---|---|---|---|---|
| Myeloblast | 0 | 0.3 | -1 | -1 | -1 | -1 | -1 | -1 | 0.2 | -1 | -1 | 0 | -1 | 0.2 | 0 |
| Promyelocyte | 0.3 | 0 | 0.3 | -1 | -1 | -1 | 0.2 | 0.2 | -1 | -1 | -1 | 0 | -1 | 0.2 | 0 |
| Myelocyte | -1 | 0.3 | 0 | 0.3 | -1 | -1 | 0.2 | 0.2 | -1 | -1 | -1 | 0 | -1 | 0.2 | 0 |
| Metamyelocyte | -1 | -1 | 0.3 | 0 | 0.3 | -1 | -1 | -1 | -1 | -1 | -1 | 0 | -1 | 0.2 | 0 |
| Band Neut. | -1 | -1 | -1 | 0.3 | 0 | 0.3 | -1 | -1 | -1 | -1 | -1 | -1 | -1 | 0.2 | 0 |
| Segm. Neut. | -1 | -1 | -1 | -1 | 0.3 | 0 | -1 | -1 | -1 | -1 | -1 | -1 | -1 | 0.2 | 0 |
| Eosinophil | -1 | 0.2 | 0.2 | -1 | -1 | -1 | 0 | -1 | -1 | -1 | -1 | -1 | -1 | 0.2 | 0 |
| Basophil | -1 | 0.2 | 0.2 | -1 | -1 | -1 | -1 | 0 | -1 | -1 | -1 | -1 | -1 | 0.2 | 0 |
| Monocyte | 0.2 | -1 | -1 | -1 | -1 | -1 | -1 | -1 | 0 | -1 | -1 | -1 | -1 | 0.2 | 0 |
| Lymphocyte | -1 | -1 | -1 | -1 | -1 | -1 | -1 | -1 | -1 | 0 | 0.2 | -1 | -1 | 0.2 | 0 |
| Plasma Cell | -1 | -1 | -1 | -1 | -1 | -1 | -1 | -1 | -1 | 0.2 | 0 | -1 | -1 | 0.2 | 0 |
| Erythroblast | 0 | 0 | 0 | 0 | -1 | -1 | -1 | -1 | -1 | -1 | -1 | 0 | 0.3 | 0.2 | 0 |
| RBC/Platelet | -1 | -1 | -1 | -1 | -1 | -1 | -1 | -1 | -1 | -1 | -1 | 0.3 | 0 | 0.2 | 0 |
| Rare/Atypical | 0.2 | 0.2 | 0.2 | 0.2 | 0.2 | 0.2 | 0.2 | 0.2 | 0.2 | 0.2 | 0.2 | 0.2 | 0.2 | 0 | 0.2 |
| Artifact | 0 | 0 | 0 | 0 | 0 | 0 | 0 | 0 | 0 | 0 | 0 | 0 | 0 | 0.2 | 0 |

## A.3 BMC Dataset ($21 \times 21$)

Table 5: Constraint matrix **C** for the BMC dataset.

|  | Myelo. | Prom. | Myelo. | Meta. | Band | Seg. | Eos. | Baso. | Mono. | Lymph. | Plas. | Eryth. | Mega. | Pro-E | Baso-E | Poly-E | Ortho-E | RBC | Art. | Smudge | Other |
|---|---|---|---|---|---|---|---|---|---|---|---|---|---|---|---|---|---|---|---|---|---|
| Myeloblast | 0 | 0.3 | -1 | -1 | -1 | -1 | -1 | -1 | 0.2 | -1 | -1 | 0 | 0 | 0 | 0 | -1 | -1 | -1 | 0 | 0.15 | 0 |
| Promyelocyte | 0.3 | 0 | 0.3 | -1 | -1 | -1 | 0.2 | 0.2 | -1 | -1 | -1 | 0 | 0 | 0 | 0 | -1 | -1 | -1 | 0 | 0 | 0 |
| Myelocyte | -1 | 0.3 | 0 | 0.3 | -1 | -1 | 0.2 | 0.2 | -1 | -1 | -1 | 0 | 0 | 0 | 0 | -1 | -1 | -1 | 0 | 0 | 0 |
| Metamyelocyte | -1 | -1 | 0.3 | 0 | 0.3 | -1 | -1 | -1 | -1 | -1 | -1 | 0 | 0 | 0 | 0 | -1 | -1 | -1 | 0 | 0 | 0 |
| Band Neut. | -1 | -1 | -1 | 0.3 | 0 | 0.3 | -1 | -1 | -1 | -1 | -1 | -1 | -1 | -1 | -1 | -1 | -1 | -1 | 0 | 0 | 0 |
| Segm. Neut. | -1 | -1 | -1 | -1 | 0.3 | 0 | -1 | -1 | -1 | -1 | -1 | -1 | -1 | -1 | -1 | -1 | -1 | -1 | 0 | 0 | 0 |
| Eosinophil | -1 | 0.2 | 0.2 | -1 | -1 | -1 | 0 | -1 | -1 | -1 | -1 | -1 | -1 | -1 | -1 | -1 | -1 | -1 | 0 | 0 | 0 |
| Basophil | -1 | 0.2 | 0.2 | -1 | -1 | -1 | -1 | 0 | -1 | -1 | -1 | -1 | -1 | -1 | -1 | -1 | -1 | -1 | 0 | 0 | 0 |
| Monocyte | 0.2 | -1 | -1 | -1 | -1 | -1 | -1 | -1 | 0 | -1 | -1 | -1 | -1 | -1 | -1 | -1 | -1 | -1 | 0 | 0 | 0 |
| Lymphocyte | -1 | -1 | -1 | -1 | -1 | -1 | -1 | -1 | -1 | 0 | 0.2 | -1 | -1 | -1 | -1 | -1 | -1 | -1 | 0 | 0.15 | 0 |
| Plasma Cell | -1 | -1 | -1 | -1 | -1 | -1 | -1 | -1 | -1 | 0.2 | 0 | -1 | -1 | -1 | -1 | -1 | -1 | -1 | 0 | 0 | 0 |
| Erythroblast | 0 | 0 | 0 | 0 | -1 | -1 | -1 | -1 | -1 | -1 | -1 | 0 | -1 | 0.3 | 0.3 | 0.3 | 0.3 | 0.3 | 0 | 0 | 0 |
| Megakaryocyte | 0 | 0 | 0 | 0 | -1 | -1 | -1 | -1 | -1 | -1 | -1 | -1 | 0 | -1 | -1 | -1 | -1 | 0.2 | 0 | 0 | 0 |
| Pro-Erythro. | 0 | 0 | 0 | 0 | -1 | -1 | -1 | -1 | -1 | -1 | -1 | 0.3 | -1 | 0 | 0.3 | -1 | -1 | -1 | 0 | 0 | 0 |
| Baso-Erythro. | 0 | 0 | 0 | 0 | -1 | -1 | -1 | -1 | -1 | -1 | -1 | 0.3 | -1 | 0.3 | 0 | 0.3 | -1 | -1 | 0 | 0 | 0 |
| Poly-Erythro. | -1 | -1 | -1 | -1 | -1 | -1 | -1 | -1 | -1 | -1 | -1 | 0.3 | -1 | -1 | 0.3 | 0 | 0.3 | -1 | 0 | 0 | 0 |
| Ortho-Erythro. | -1 | -1 | -1 | -1 | -1 | -1 | -1 | -1 | -1 | -1 | -1 | 0.3 | -1 | -1 | -1 | 0.3 | 0 | 0.3 | 0 | 0 | 0 |
| RBC | -1 | -1 | -1 | -1 | -1 | -1 | -1 | -1 | -1 | -1 | -1 | 0.3 | 0.2 | -1 | -1 | -1 | 0.3 | 0 | 0 | 0 | 0 |
| Artifact | 0 | 0 | 0 | 0 | 0 | 0 | 0 | 0 | 0 | 0 | 0 | 0 | 0 | 0 | 0 | 0 | 0 | 0 | 0 | 0.2 | 0.2 |
| Smudge | 0.15 | 0 | 0 | 0 | 0 | 0 | 0 | 0 | 0 | 0.15 | 0 | 0 | 0 | 0 | 0 | 0 | 0 | 0 | 0.2 | 0 | 0.2 |
| Other | 0 | 0 | 0 | 0 | 0 | 0 | 0 | 0 | 0 | 0 | 0 | 0 | 0 | 0 | 0 | 0 | 0 | 0 | 0.2 | 0.2 | 0 |

Key relationships encoded in these matrices include: (i) sequential maturation stages have positive co-occurrence (0.3), (ii) different lineages (myeloid, lymphoid, erythroid) are mutually exclusive (-1), (iii) erythroid maturation shows strong sequential relationships (0.3), (iv) artifacts and smudge cells show weak correlations (0.15-0.2) with specific cell types and (v) early precursors can weakly co-occur with multiple lineages (0-0.2).

## Appendix B. Details about the GR-Neutro dataset

The **GR-Neutro** dataset is composed of 7 different classes including normal neutrophils, nuclear chromatin condensation, Döhle bodies (basophilic cytoplasmic inclusions), hypergranulation (increased cytoplasmic granulation), hypersegmentation (increased nuclear lobes), hypogranulation (decreased cytoplasmic granulation), and hyposegmentation (decreased nuclear lobes). Table 6 includes the number of each class for the training and testing splits.

| Split | Normal | Chromatin | Dohle | Hypergr. | Hyperseg. | Hypogr. | Hyposeg. |
|-------|--------|-----------|-------|----------|-----------|---------|----------|
| # Train | 658 | 277 | 56 | 45 | 46 | 279 | 253 |
| # Test | 220 | 93 | 19 | 15 | 16 | 93 | 84 |

Table 6: Distribution of samples across training and test splits in the GR-Neutro dataset.

## Appendix C. Cross validation results for the GR-Neutro dataset

We performed comprehensive 5-fold cross-validation to the **GR-Neutro** dataset to ensure robust evaluation of our approach. Table 7 presents detailed results across all folds. Results show consistent performance improvements across all folds, with low variance in key metrics.

Table 7: 5-Fold Cross-Validation Results

| Metric | Fold 1 | Fold 2 | Fold 3 | Fold 4 | Fold 5 |
|--------|--------|--------|--------|--------|--------|
| Accuracy | 0.93 | 0.94 | 0.92 | 0.95 | 0.93 |
| Macro F1 | 0.92 | 0.93 | 0.91 | 0.94 | 0.92 |
| AUC-ROC | 0.94 | 0.95 | 0.93 | 0.96 | 0.94 |
| **Per-Class F1-Scores** | | | | | |
| Normal | 0.98 | 0.99 | 0.98 | 0.99 | 0.98 |
| Chromatin | 0.86 | 0.88 | 0.85 | 0.89 | 0.87 |
| Döhle | 0.94 | 0.96 | 0.93 | 0.97 | 0.95 |
| Hypergran. | 0.99 | 1.00 | 0.98 | 1.00 | 0.99 |
| Hyperseg. | 0.96 | 0.98 | 0.95 | 0.98 | 0.97 |
| Hypogran. | 0.89 | 0.91 | 0.88 | 0.92 | 0.90 |
| Hyposeg. | 0.89 | 0.91 | 0.88 | 0.92 | 0.90 |

## Appendix D. Attention Heatmap Analysis

To provide insights into how our model focuses on different morphological features, we visualized the attention heatmaps for various neutrophil abnormalities. Figure 4 shows the attention patterns across different cell types, demonstrating how the model learns to focus on relevant morphological features for each abnormality type.

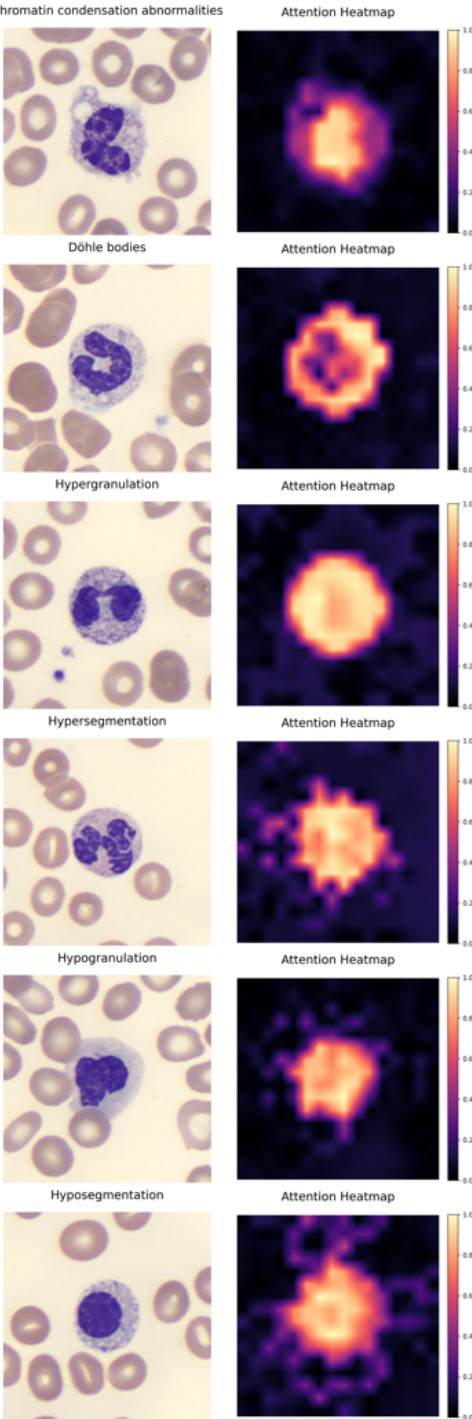

Figure 4: Attention heatmaps showing model focus areas for different neutrophil abnormalities. The visualization demonstrates how the model attends to specific morphological features characteristic of each abnormality type.

