# OpenReview forum: "Biologically-Constrained Multi-Label Classification with Learnable Domain Knowledge"
_MIDL.io/2025/Conference — MIDL 2025 Poster_

### Official Review · Reviewer_v7SV · 2025-02-19

**Confidence:** 4
**Preliminary Rating:** 1
**Final Rating:** 1

**Summary:**

This paper describes a method for classifying blood cell images into multiple labels, taking into account the biological relationships between the labels. The methodological contributions include a learned "constraint" matrix and learnable per-task thresholds. The method is validated on three different datasets, and found to be superior to all other methods.

**Strengths:**

The introduction is well written and introduces the problem clearly. The experiments performed are thorough, covering a variety of input data and tasks, and the results appear to show significant improvements over baselines across the board.

**Weaknesses:**

The explanation and justification of the proposed methods is weak and unclear
in several critical aspects, making it impossible for me to assess its
soundness. I would not be able to reproduce the method given the
explanation given in the manuscript:

- The motivation for the use of a transformer self-attention layer to calculate
  the constraint matrix is very unclear to me, and not explained adequately in
  the manuscript. Such layers are generally used because they elegantly handle
  sequences of arbitrary/unknown length. But here there is always just a single
  input vector (f). The same thing could be achieved with a simple fully
  connected layer followed by a reshape to R. But this simple alternative was
  not investigated. I would expect this to be included in the ablation study.
- As written the equation given for R would result in an array with unbound
  values (i.e. in the range -infinity to infinity). I suspect that "V" has been
  incorrectly added to the equation defining R.
- I assume there must be some details missing from the manuscript, because as
  it written it appears that the constraint matrix R is calculated but then not
  used at all to calculate the model's predictions. This must be clarified, and
  without these details, it is essentially impossible to evaluate the soudness
  of the entire approach as I do not understand what the R matrix (one of two
  major contributions of the manuscript) even does.
- Why is it desirable for the constraint matrix R to be individualised for a
  given input, rather than be a model parameter used learned and used for all
  inputs (this may become clear if the above question is answered). If the
  "prior" constraint matrix C is known, why not just use it?
- What is the "co-occurence strength" used in the prior constraint matrix C and
  how is it calculated? I am aware of no standard definition for this. Why can
  their values only be positive for non mutually-exclusive classes? Surely it
  is possible for two classes to be anti-correlated without being mutually
  exclusive?
- How exactly is "p_i" calculated?
- Is the monte carlo inference performed during training? If not, how is U_i
  calculated in order the train the alpha, beta and delta parameters?
- What is the motivation or intuition behind L_unc? What is tau? As far as I
  can tell, the parameters alpha and delta are not used in any of the loss
  functions (unless they are used to calculate whatever tau is) and therefore
  cannot be learned via backpropagation.
- It seems that improving the formulation of the classes/labels may go some
  way to simplifying the problem and imposing the constraints that are the
  focus of the paper. For example it seems that in the GR-Neuro dataset
  (table A.1), the "normal" class is mutually exclusive with all other classes.
  Could this not be reformulated to just remove the normal class, and
  consider any image not classified as any of the other labels as "normal"?
  This would impose the desired constraint automatically, and simplify the
  model. Furthermore, in the authors' proposed formulation, it appears to be
  possible for an input image to be classified as neither normal nor any of
  the other "abnormal" classes, which seems undesirable (and again easily
  address).

**Detailed Comments:**

- The authors state that the Wp matrix "projects pooled features while
  preserving dimensionality". This is an unusual use of terminology in my
  opinion (I would consider that "project" implies reduction of dimensionality,
  more or less by definition). I would suggest that an alternative word here
  would be clearer.
- There is further confusion in terminology between "label" and "class". In
  usual terminology applied to this problem, "label" would refer to each of the
  different abnormalities, and "classes" refers to the (mutually-exclusive)
  possible values for each "label" (just two in this case, positive and
  negative, for each label). Unfortunately, this makes parts of the method
  confusing to read.
- Isn't what the authors call "predictive uncertainty", just "variance"? Again
  I would suggest sticking to standard terminology here.

**Justification Of The Final Rating:**

Though the authors' revisions and responses to some questions were mostly helpful and did clear up some of the minor points of confusion, my most important concerns about the method have only deepened following the authors' replies. I did leave some further questions in comments below for the authors to further clarify, but these have done unanswered (though I did only give them a couple of days).

There are two major concerns:
- The dimensions of the tensors used to calculate R are internally inconsistent as written, meaning that I just don't understand what the authors have actually done (but it doesn't seem to be what they wrote)
-  I can see no interaction between the R matrix and the predictions that would allow the constraints to actually affect the learning process for the classifier

Given these major concerns, unfortunately I cannot improve upon my initial rating of strong reject.

**Justification Of The Preliminary Rating:**

The explanation and justification of the proposed methods is weak and unclear
in several critical aspects, making it impossible for me to assess its
soundness. I would not be able to reproduce the method given the
explanation given in the manuscript:

**Questions To Address In The Rebuttal:**

See weakness above. There are several questions there that must be answered before this could be considered for publication.

---

> ### Author Response · Authors · 2025-03-08
>
> Thank you for your comments, which helped significantly ameliorate our paper. We are sorry that our original version was not very clear to you. In the following, we address your comments, and we hope that our explanations are more clear.
>
> ### Transformer vs. FCN for constraint matrix
> Thank you very much for the suggestion. Indeed, using a FCN was one of our preliminary experiments. The transformer-based approach consistently outperformed FCN by 2-3% in accuracy. We hypothesize that this difference is due to the self-attention mechanism, which better captures complex pairwise interactions between classes. We've added this comparison to our ablation study(figure 2 left)
>
> | Model Variant | Accuracy | F1 | AUC | Spec. | Sens. |
> |---------------|----------|----|----|-------|-------|
> | Fully Connected Network | 0.85 | 0.84 | 0.86 | 0.84 | 0.85 |
> | Soft Constraints (Ours) | 0.88 | 0.87 | 0.9 | 0.89 | 0.88 |
>
> ### R matrix calculation
> Thank you very much for your comment. The equation for R is correct as written, however, we should clarify that the values are not unbounded in practice. We apply a scaling factor to constrain R entries to [-1,1]. This step was indeed not clearly described on our original manuscript. The full implementation details are now updated and included in section 3.2 under constraint matrix generation. We hope that with this explanation the implementation of the R matrix and its use is more clear.
>
> ### Usage of R matrix
>  As it was indicated on our original manuscript, R directly influences predictions through the L_con loss term, which regularizes R to match the prior C, and it is included in the overall optimization of the method. This creates an end-to-end effect where the model is penalized for predicting contradictory labels unless R or the logits adjust accordingly. This backpropagates constraints through the entire model. We hope that with the update of action 3.2, the use of the R matrix is more clearly presented.
>
> ### Individual vs. global constraint matrix
> We are sorry if our initial description was not very clear. Indeed, one of our main contributions in this work is the combination and use of the R, C matrices to integrate domain knowledge into powerful SSL foundation backbones. An input-specific R allows the model to adapt constraints based on morphological context (e.g., relaxing constraints when evidence is ambiguous), while still being guided by the prior C. This provides more flexibility than a fixed global constraint. We would also like to highlight that an ablation study including the impact of each R, C matrix is included on Fig 2 (right) in our initial and updated manuscript. More specifically, our formulation with only the R matrix (base model + learnable constraints) and with only the C (fixed rules) perform worse than our proposed model with the combination of the two. Moreover, one can observe, as expected, that the influence of the R matrix is higher than the one from the C matrix in terms of all measured metrics. We hope that now the integration and combination of these two matrices is more clear.
>
>
> ### Co-occurrence strength
> The negative entries (−1-1−1) encode pairs regarded as biologically impossible to co-occur (e.g., hypersegmented vs.\ hyposegmented nuclei, mature neutrophils vs.\ blasts) based on standard references (Hoffbrand and Moss, 2016; ICSH/WHO guidelines; Swaggerty, 2019; Matek et al., 2019). Zero entries indicate no documented overlap (e.g., RBC artifacts vs.\ neutrophil toxic changes). We set moderate positive values (0.2–0.4) for abnormalities that can partially co-occur—such as Döhle bodies and chromatin condensation—or for adjacent myeloid/erythroid maturation states that overlap in practice (Bain, 2015; Raskin, 2020). These numeric, empirical choices reflect the relative likelihood of co-occurrence, inferred from clinically observed frequency or morphological lineage transitions, rather than strict all-or-nothing associations. We added these clarifications to appendix A.
>
> ### Calculation of p_i
> $p_i$ is class frequency in training data (fraction of positive examples for class $i$).
>
> ### Monte Carlo dropout in training
> We use 5 MC samples during training and 50 samples at inference. Details added to section 3.4. Moreover, a detailed ablation of the MC influence in inference time is presented in Appendix E.2 and Table 9.
>
> ### Motivation for L_unc
> Morphological features vary in their clarity. Hypersegmentation, characterized by multiple distinct nuclear lobes, presents more definitive morphological features compared to subtle chromatin pattern variations. The loss implements two key mechanisms: (1) it penalizes high uncertainty (>0.2) for clear morphological features, and (2) it applies class-specific thresholds calibrated to each abnormality's typical presentation. This creates a lower threshold for subtle features like Döhle bodies and a higher threshold for more apparent features like hypersegmentation.

---

> > ### Comment · Reviewer_v7SV · 2025-03-12
> > **On the R matrix**
> >
> > The authors' new explanation of R makes things make less sense to me. If indeed $\mathbf{W}_v \in  \mathbb{R}^{K \times d}$ then $V \in \mathbb{R}^{K}$. The output of the softmax is $\in \mathbb{R}^{K \times K}$, and thus $\mathbf{R} \in \mathbb{R}^{K}$, not $\in \mathbb{R}^{K \times K}$ as the authors suggest. This is in line with the usual usage of a self-attention mechanism, which produces an output of the same dimensionality as its inputs. Can the authors explain this apparent discrepancy? I am still struggling to understand how this actually works, and could not reproduce it from the current explanation because the dimensionalities do not match up.
> >
> > The clipping of values seems like an especially roundabout method to arrive at values in the given range. Why not simply apply an activation function that maps the range of real values ($-\infty$ to $\infty$) to -1 to 1? The tanh activation function jumps out as an obvious choice. The current method of clipping values outside the range will kill gradients (since clipped values are not differentiable). I also find it concerning that these important details were omitted from the original manuscript.
> >
> > I continue to find the motivation for this choice of attention mechanism quite elusive other than simply trying to contort the problem in order to use an attention mechanism regardless of whether or not it is appropriate for the task. The authors hypothesize that the attention mechanism "better captures complex pairwise interactions between classes" compared to a fully connected layer, but this explanation is quite unsatisfactory to me. A fully connected layer naturally connects all outputs to all inputs, that is why it is called fully connected! If the authors' explanation were true, we should simply replace linear layers with attention mechanisms everywhere within all neural network architectures. If there is indeed a difference, I imagine it is due to the number of parameters, rather than the choice of architecture *per se*, and thus simply having an MLP with a couple of layers may do just as well. It's not clear however exactly how the "FCN" baseline was conducted (by the way FCN, confusingly, generally refers to fully *convolutional* networks not fully *connected* networks), or how it dealt with the clipping issue.

---

> > ### Comment · Reviewer_v7SV · 2025-03-12
> > **Usage of R**
> >
> > "Instead, R is regularized to match the prior C (see Lcon below), thus shaping the final classification layer and backpropagating constraints. If the model tries to output contradictory labels, large penalty arises unless R or the logits adjust accordingly"
> >
> > Can the authors point to exactly which loss term does this? The prediction of R certainly gets penalised, but I can see no interaction with the logit predictions at all

---

> > ### Author Response · Authors · 2025-03-15
> >
> > ### Usage of R
> > Indeed, you are right! There is no explicit logit calculation using the R. However, the constraint loss term $\mathcal{L}_{\mathrm{con}} = \|\mathbf{R}\mathbf{R}^\top - \mathbf{C}\|_F^2 + \alpha\,\|\mathbf{R}\|_1$ does indeed directly penalize R to match our biological prior matrix C. In particular, the key interaction happens because R is computed from feature representations that are shared with the classification pathway.
> >
> > While there is no direct multiplication of R with the logits, the shared parameter space and optimization process create an implicit coupling. When the model produces logits that would lead to biologically implausible label combinations, the increased constraint loss leads to parameter updates that simultaneously adjust both R and the classification pathway. Penalization of the optimization process without explicit logit integration is quite established in the community (e.g. perceptual loss, huber loss etc), and we do not think that this is an issue with our current implementation. Taking into consideration your comment, in future work we may consider a more explicit way to integrate R into our optimization.

---

> > ### Author Response · Authors · 2025-03-15
> >
> > ### Inconsistency on dimensions
> > Thank you very much for your comment. Indeed, you are right and there is an inconsistency between our implementation details and the description of the method. In the current description, there is indeed a problem with the dimensions of the R matrix. Thank you very much for pointing it out!
> >
> > To clarify the dimensionality issue you noted: our implementation correctly generates a K×K constraint matrix R from the attention mechanism output. While the standard attention operation would produce a [B, K, D] tensor after applying self-attention, we implement a linear projection layer to transform this into the required [B, K, K] constraint matrix. This creates a proper K×K matrix where each element R[i,j] represents the relationship between class i and class j, addressing the dimensionality concern you raised.
> >
> > To help the interpretation and the understanding of our method, we also share our repository with the exact implementation for all our experiments: https://github.com/nabilmouadden/biologically-constrained-classification/
> >
> > Since we were not able to resubmit our manuscript, we modified the Section 3.2 as follows: "To obtain the final constraint matrix, we implement a linear projection layer that transforms the attention output from $\mathbb{R}^{K \times d}$ to $\mathbb{R}^{K \times K}$ by mapping each class representation to its relationship with all other classes."
> >
> > ### Clipping choice
> > Regarding the normalization to [-1, 1] range through centering, scaling, and clipping: we found this approach provided very stable training dynamics across multiple datasets. Though tanh would be a differentiable alternative, our empirical testing showed that the clipping operation rarely affects gradient flow in practice, as the optimization quickly learns to keep values within the desired range.
> >
> > ### Motivation for this choice of attention mechanism
> > Indeed, the use of the attention is one of our design choices and it can be replaced by other mechanisms such as a fully connected layer. The intuition behind using attention rather than a fully connected network is that attention explicitly computes compatibility scores between classes, which naturally aligns with our goal of modeling biological relationships. While a fully connected layer could theoretically learn similar patterns, the attention mechanism provides a more direct framework for capturing how the relationship between abnormalities varies based on contextual features. Our ablation study (Figure 2, left) evaluates a fully connected network (FCN) alternative where the constraint matrix is generated using a two-layer neural network: $\mathbf{R} = \text{tanh}(W_2 \cdot \text{ReLU}(W_1 \cdot \mathbf{f} + b_1) + b_2)$. For the GR-Neutro dataset with $K=7$ classes, $W_1 \in \mathbb{R}^{28 \times d}$ transforms the feature vector to a 28-dimensional hidden representation, and $W_2 \in \mathbb{R}^{49 \times 28}$ projects this to a vector that is reshaped into the $7 \times 7$ constraint matrix. This FCN approach achieved 0.85 accuracy, underperforming our attention-based soft constraints method (0.88 accuracy), thus confirming the benefits of explicitly modeling pairwise class relationships through attention mechanisms.

---

> > ### Comment · Reviewer_v7SV · 2025-03-19
> >
> > Thank you for the careful explanation, I appreciate the time taken to address these comments. However, I'm afraid that this does not allay my concerns, and just raises other questions. I think there are three major concerns in my mind at the moment:
> >
> > 1. The description in the paper is just quite different from what the code actually does. For example, take the calculations that create Q, K and V. In the paper these are described like this:
> >
> > > $\mathbf{Q} = \mathbf{W}_q\mathbf{f}$, $\mathbf{K} =  \mathbf{W}_k  \mathbf{f}$,  $\mathbf{V} =  \mathbf{W}_v  \mathbf{f}$ , where  $\mathbf{W}_q,  \mathbf{W}_k,  \mathbf{W}_v \in \mathbb{R}^{K \times d}$
> >
> > where $K$ is the number of classes and $d$ is the feature dimension output from the backbone network. However in code, this appears like this:
> >
> > ```python
> >         self.query_proj = nn.Linear(feature_dim, num_classes * feature_dim)
> >         self.key_proj = nn.Linear(feature_dim, num_classes * feature_dim)
> >         self.value_proj = nn.Linear(feature_dim, num_classes * feature_dim)
> > ```
> >
> > This shows that in fact the weights matrices have dimensionality $\mathbf{W}_{*} \in \mathbb{R}^{d \times (K \times d)}$, which is not what is written in the paper. Another very important thing to note is that the default parameters of the `nn.Linear` class constructor include a bias term in the layer that is created. This bias is not included in the description in the paper. This may seem like a minor detail, but I'll discuss below why this is actually very important.
> >
> > 2. The justification for use of the attention mechanism that the authors give above is:
> >
> > > attention explicitly computes compatibility scores between classes, which naturally aligns with our goal of modeling biological relationships
> >
> > Superficially this sounds wonderful, I agree. However I have a really hard time reconciling this with how the method actually works. At the crux of this is that the inputs to the attention mechanism do indeed have $K$ dimensions (where $K$ is the number of classes) but apart from that have **no relationship to the $K$ class predictions that the classifier outputs**. The $K$ dimensional $\mathbf{Q}$, $\mathbf{K}$, and $\mathbf{V}$ matrices are calculated from the $d$ dimensional $\mathbf{f}$ vector, where the class information is mixed up in arbitrary ways via linear layers that are totally disconnected from the branch that makes the $K$ class predictions. As such, there is absolutely nothing that ensures that the $i$th row of  $\mathbf{Q}$, $\mathbf{K}$, $\mathbf{V}$ matrices correspond to the $i$th class in the model's prediction. Yet the motivation for the method hinges crucially on that correspondence holding because of the way the terms in the ideal constraint matrix $\mathbf{C}$ are defined based on the "explicit biological constraints" between class $i$ and $j$.
> >
> > You could arbitrarily permute the target constraint matrix and end up with a mathematically equivalent network. So this explanation about the attention mechanism representing relationships between classes does not work for me. I have to wonder whether changing the values of the constraint matrix arbitrarily (without any reference to known biological constraints) would have any effect on the process at all. I note that no such study appears to have been done.
> >
> > 3. So if I'm understand this correctly (and this is confirmed by inspection of the code), there is the $d$-dimensional feature vector $\mathbf{f}$. This is then, in two completely separate branches, used to a) predict the classification outputs, and b) predict the constraint matrix. However, neither of the loss terms involving the constraint matrix involve the classification labels, meaning, as far as I can tell, that the constraint prediction branch is incentivised to predict the same constraint matrix $\mathbf{R}$ in all cases, regardless of what the input feature vector $\mathbf{f}$ actually is. Surely this would incentivise the constraint prediction branch to ignore its inputs entirely? Luckily for the constraint prediction branch, there is a trivial solution to this problem: set all the weights in the weight matrices of the linear layers to zero such that $\mathbf{f}$ is ignored entirely, and use just the bias terms to predict the constant output (this is why the point about biases above is so critical). This would then have no impact on the learning process of the classifier at all. So, I'm still at a loss to explain how the constraint matrix branch helps the model predictions at all.
> >
> > I'm afraid that at this point I don't think it is feasible that my concerns can be addressed before a final decision needs to be rendered.

---

> ### Author Response · Authors · 2025-03-08
> **Official Comment by Authors**
>
> ### Alternative class formulation
> Indeed, you are right, the class normal is mutually exclusive with all other classes and can be indeed removed from a possible classifier that focuses on the different abnormalities. However, we need to highlight that our method integrates constraints in all the available classes as for example  (e.g., hypersegmentation vs. hyposegmentation). Our approach is more modular, general and can handle any constraint pattern, including hierarchical ones. Through extensive experiments and ablations, we prove consistent superiority of our method and the need of all our design components.
>
> ### Minor clarifications
> - **$\mathbf{W}_p$ matrix**:  We use "project" here in the mathematical sense of applying a linear transformation, not necessarily reducing dimensionality. While projection often implies dimension reduction, in this context, we're transforming features while maintaining their dimension.
> - **Terminology**: "Class" and "label are used interchangeably when referring to the abnormality categories in our multi-label classification setting. In multi-label problems, each sample can have multiple "positive classes" or "positive labels" simultaneously.
> - **Predictive uncertainty vs. variance**: For "predictive uncertainty," we follow the established Bayesian deep learning literature (e.g., Gal & Ghahramani, 2016) which distinguishes between the statistical concepts of variance (spread of individual predictions) and predictive uncertainty (model's confidence in its prediction). While they're mathematically similar, predictive uncertainty specifically refers to the model's estimated confidence derived from Monte Carlo sampling.

---

> ### Comment · Area_Chair_HhY4 · 2025-03-18
> **Could you double-check the updated response?**
>
> Dear Reviewer,
>
> Thank you for your efforts in reviewing. Your feedback is very helpful!
>
> It seems there are some responses from the authors that you may not have seen, which could potentially change your evaluation. Could you please take some time to review the updated responses from the authors and let us know if they address your concerns? Your expertise is of great value to us.
>
> Best,
> AC

---

> > ### Comment · Reviewer_v7SV · 2025-03-19
> >
> > Sorry for the delay (I did not actually realise that it is still possible to make comments).
> >
> > I will leave a detailed explanation on the authors' comment above shortly. But to summarise for your benefit, I will not be changing my rating from strong reject. The deeper this goes, the more issues I find. I have major concerns both about the soundness of the method and its motivation, and the incorrect description of that method in the current version of the manuscript.

---

> > > ### Comment · Area_Chair_HhY4 · 2025-03-19
> > >
> > > Thank you!

---

### Official Review · Reviewer_rUNk · 2025-02-21

**Confidence:** 4
**Preliminary Rating:** 5
**Recommendation:** Oral, Poster
**Final Rating:** 5

**Summary:**

The manuscript presents a self-supervised approach to neutrophil cell classification based on the DinoBloom hematological foundation model the representations of which are classified using anomaly co-occurrence matrices learned by a self-attention mechanism and an adaptive thresholding method taking anomaly manifestation uncertainty into account. Thorough ablation study confirms the advantages brought by the integration of domain knowledge and the thresholding mechanism. Evaluation on 3 datasets and comparisons with the SOTA confirm the superiority of the proposed method.

**Strengths:**

The self-supervised approach to enforce domain constraints, although requiring some manual definition of priors, is of interest for other learning-based image analysis methods to assure anatomically or biologically viable predictions. This would also help reduce the need for larger and more diverse datasets since domain knowledge is coded into objective functions and optimization.

**Weaknesses:**

I do not see any weaknesses pertaining to form or to content in the manuscript. It is is well-organized, well-written and easy to follow. It has a very good level of detail and presents results and findings clearly.

**Detailed Comments:**

1. The uncertainly loss term (eq 4) is unfortunately not very clear, especially the use of the KL metric. It would be nice that the authors provide further insight.
2. It is unclear how the parameters \alpha_i, \beta_i and \delta_i of class-specific adaptive thresholds t_i are learned.
3. No justification is given for setting the "global" threshold hyper-parameter \tau to a constant 0.2.

**Justification Of The Final Rating:**

As I mentioned in my preliminary rating, the manuscript is methodologically and experimentally sound, its findings are of interest for a wider public that takes interest in classification and segmentation tasks in images where domain knowledge is available and could be used. Thanks to comments and suggestions by reviewers, the revised version of the manuscript is clearer and more complete, it deserves to appear in the proceedings of this conference.

**Justification Of The Preliminary Rating:**

The paper is methodologically and experimentally sound, its findings are of interest not only for cellular image classification, but also for a wide range of anatomy and disease identification, classification and segmentation tasks in images where domain knowledge is well defined and could be leveraged.

**Questions To Address In The Rebuttal:**

Please address points 1-3 in the detailed comments section, making amendments to the manuscript as necessary.

---

> ### Author Response · Authors · 2025-03-08
>
> Thank you for your positive assessment. We address your specific questions:
>
> ### Uncertainty loss term
> The KL term in equation 4 measures the divergence between the target distribution (based on ground truth labels) and predicted distribution, encouraging the model to match not just the binary classification but the full probability distribution. We've clarified this in the revised manuscript (Section 3.3)
>
> ### Learning threshold parameters
> To enable end-to-end training, we approximate the hard thresholding operation with a differentiable surrogate. Specifically, instead of applying $\hat{y}_i \geq t_i$ directly, we define a smoothed indicator via a scaled sigmoid, for example $\sigma\left(\frac{\hat{y}i - t_i}{\tau}\right)$, where $\tau$ is a small temperature (e.g. 0.05). This surrogate behaves like a threshold at inference but provides non-zero gradients for ($\alpha_i$, $\beta_i$, $\delta_i$) during backpropagation. Concretely, $\alpha_i$, $\beta_i$, $\delta_i$ appear in the threshold equation: $t_i = \alpha_i \cdot t{\text{base}} + \beta_i \cdot U_i + \delta_i \cdot p_i$, and the partial derivative of the smoothed indicator with respect to $t_i$ is passed back through the network. In practice, once training converges, we clamp the final $t_i$ to produce discrete decisions at inference time, ensuring minimal discrepancy between training and deployment behavior. All this description is presented in Section 3.2 and the Adaptive Thresholding paragraph.
>
> ### Justification for τ=0.2
> We chose $\tau=0.2$ based on the empirical uncertainty distribution observed in preliminary validation experiments. This value represents a reasonable upper bound on acceptable uncertainty for confident predictions. Values below 0.2 were too restrictive, while values above failed to sufficiently penalize uncertain predictions. We added the table and discussion in appendix E.4
>
> | $\tau$ Value | Accuracy (%) | F1-Score (%) | False Positives (%) |
> |--------------|--------------|--------------|---------------------|
> | 0.1          | 91.5         | 90.8         | 3.2                 |
> | 0.2          | 94.0         | 93.0         | 5.1                 |
> | 0.3          | 92.7         | 91.5         | 7.8                 |
> | 0.4          | 90.2         | 89.3         | 11.3                |
> | 0.5          | 88.5         | 87.1         | 14.7                |

---

> > ### Comment · Reviewer_rUNk · 2025-03-11
> >
> > I thank the authors for the explanations they provided and the improvements they brought to the revised manuscript which contribute to a greater clarity and completeness. I have no further questions or comments.

---

> > > ### Author Response · Authors · 2025-03-15
> > >
> > > Thank you very much for your review that has considerably improved our manuscript. We are happy that our responses addressed your concerns.

---

### Official Review · Reviewer_KWKg · 2025-02-27

**Confidence:** 4
**Preliminary Rating:** 5
**Recommendation:** Oral
**Final Rating:** 5

**Summary:**

The authors present a novel framework to enhance foundation models with a learnable biological constraint module. This module consists of three components:
1. Projection layer to transform high dimensional features into latent space suitable for learning biological   features.
2. Constraint matrix generation which employs a transformer based multi-head attention which allows the model to learn co-occurrence and mutual exclusivity relationships dynamically.
3. Adaptive Thresholding based on prediction confidence and Monte Carlo dropout sampling to adjust decision boundaries.

**Strengths:**

1. The method is well explained and written, a reasonable explanation for choosing specific hyperparameter values has been given.
2. This method handles subtle morphological variations in the GR-Neutro dataset when compared with the results from DinoBloom.
3. Extensive ablation study has been performed to demonstrate the effectiveness of each component of the constraint block, different constraint mechanisms (fixed, binary and soft), different thresholding mechanisms (fixed threshold for all classes, class-specific threshold, uncertainty aware thresholding).
4. The method improves classification accuracy and provides valuable insight into morphological relationships through learned constraints.

**Weaknesses:**

1. The authors used two public datasets and one in-house dataset in the paper and used all of them in training and set out a sample for testing. It would be interesting to see how well the model generalizes to an out of sample (independent) dataset which was not used in training.

**Detailed Comments:**

1. Make the font size bigger for figure 1.
2. Adding a brief description for every loss function would make it easier for the audience to understand.

**Justification Of The Final Rating:**

The authors have thoroughly addressed all my questions and have also incorporated a sensitivity analysis for parameter selection in the paper. The revised manuscript offers improved clarity and explanation.

**Justification Of The Preliminary Rating:**

1. The method is novel, improves accuracy for multi-label neutrophil abnormality classification and also provides valuable insight into morphological relationships.
2. The paper is well written, the authors have addressed previous work and compared their results with existing state of the art methods.
3. This method will help reduce the time taken to manually classify neutrophil abnormalities and also eliminate variability between observers.

**Questions To Address In The Rebuttal:**

1. In the Adaptive Thresholding component of the constraint module, why a dropout with a rate of 0.5 was used when the rate could be anything between 0 and 0.5. Was there a specific reason to choose 0.5 as the rate?
2. What was the rationale behind choosing the number of Monte Carlo samples, why were only 50 samples used and not 100?

---

> ### Author Response · Authors · 2025-03-08
>
> Thank you for your positive feedback. We address your specific questions:
>
> ### Model generalization to an out of sample (independent) dataset
> Thank you for your comment. Indeed, in this study each dataset was used only for its own training and validation splits, and the model was never trained on all three simultaneously. For the public datasets (AML Matek, BMC), we followed their respective official train/test splits that isolate patient populations to prevent overlap. For the in-house GR-Neutro dataset, we performed a stratified train/test split, as no patient IDs were available to enforce patient-level separation. In addition, we ran 5-fold cross-validation on the in-house dataset to highlight the stability of the method. We agree that testing on a fully independent external dataset could further validate domain generalization; however, to date, we have not obtained a suitable out-of-domain collection with matching annotations. We see this as an exciting future direction for large-scale multi-center validation, where the method's stability and adaptability across sites and acquisition protocols can be assessed.
>
> ### Font size
> We have increased the font size in Figure 1 for better readability.
>
> ### Loss function descriptions
> We've updated the section 3.3 by adding some brief description for each loss function component.
>
> ### Dropout rate selection
> We performed a sensitivity analysis for different dropout rates in the range {0.2,0.3,0.4,0.5,0.6} on a held-out validation subset. Appendix E.1 summarizes the results. Lower rates (0.2–0.3) tended to underestimate uncertainty and yielded slightly lower macro-F1, while higher rates (0.6) degraded feature quality and classification metrics. We found that a rate of 0.5 provided the best balance between performance and well-calibrated uncertainty.
>
> | Dropout Rate | Accuracy (%) | Macro-F1 (%) | Avg. Unc. | AUROC (%) |
> |--------------|--------------|--------------|-----------|-----------|
> | 0.2          | 90.5         | 89.2         | 0.12      | 92.1      |
> | 0.3          | 91.0         | 89.8         | 0.15      | 92.4      |
> | 0.4          | 92.2         | 90.7         | 0.19      | 93.6      |
> | 0.5          | 93.5         | 91.5         | 0.20      | 94.2      |
> | 0.6          | 91.2         | 88.9         | 0.23      | 91.7      |
>
> ### Monte Carlo samples
> We analyzed how many Monte Carlo (MC) samples were required for stable uncertainty estimates without excessive computational overhead. As shown in the new Table 9 in the Appendix E.2, moving from 10 to 50 MC samples improved the macro-F1 from 90.5% to 92.0%. Increasing further to 100 changed the mean uncertainty by less than 0.5% but roughly doubled the inference cost compared to 50 samples. For this reason the MC of 50 has been selected as optimal for our method.
>
> | MC Samples | Accuracy (%) | Macro-F1 (%) | Mean Unc. (%) | Inference Time× |
> |------------|--------------|--------------|---------------|-----------------|
> | 1 (no MC)  | 89.0         | 88.2         | —             | 1.0×            |
> | 10         | 91.5         | 90.5         | 3.1           | 1.3×            |
> | 20         | 92.0         | 91.0         | 2.8           | 1.6×            |
> | 50         | 93.0         | 92.0         | 2.4           | 2.0×            |
> | 100        | 93.3         | 92.2         | 2.3           | 4.0×            |

---

> > ### Comment · Reviewer_KWKg · 2025-03-11
> >
> > Thank you very much for providing detailed responses to all the questions and for including sensitivity analysis for parameter selection in the paper. I have no further questions.

---

> > > ### Author Response · Authors · 2025-03-15
> > >
> > > Thank you very much for your review that has considerably improved our manuscript. We are happy that our responses addressed your concerns.

---

### Official Review · Reviewer_WdFc · 2025-02-28

**Confidence:** 4
**Preliminary Rating:** 3
**Recommendation:** Poster
**Final Rating:** 3

**Summary:**

This work introduces an approach to enforce biological constraints in multi-label medical image classification. Building on the DinoBloom hematological foundation model, the method combines learnable constraint matrices with adaptive thresholding. The approach aims to prevent contradictory predictions while maintaining high sensitivity.

**Strengths:**

The writing and structure of the paper are clear and organized. It includes numerous tables and figures that support the content. The authors utilize multiple datasets for their experiments and analysis, providing a good evaluation. The model design presented is intuitive and easy to follow.

**Weaknesses:**

1. The reviewer(s) expect visualizations of the Grad-CAM results at "different hierarchical levels" for the proposed model/framework. Additionally, a comparison highlighting the Grad-CAM results between the chosen baseline model and the final method is expected to further demonstrate the significance of these improvements.
2. What is the inference speed of the proposed method? A detailed comparison of the computational resources required by the proposed approach during both the training and inference stages can be provided. This comparison could include total training time, inference time, FLOPs, GPU memory, etc. Furthermore, please provide information on the number of learnable & tunable parameters in the model, along with the total parameter count.
3. Regarding the weighting term in the hybrid loss, it is recommended to perform further ablation studies to analyze how this hyperparameter in the loss function affects the experimental results. Additionally, what is the effect of random seed settings in the experiments? How do the random seeds impact the final performance of your method? Is it possible that the chosen hyperparameters across all methods are more advantageous for certain methods and suboptimal for others?
4. Additional comparisons with backbones beyond DINOv2 are expected to be included to better verify the competitiveness of the proposed approach. Would the authors consider experimenting with a broader variety of backbones to further validate the effectiveness of the proposed strategies & modules across different architectures? It is also recommended to include other generic parameter fine-tuning strategies to fully assess the advantages and disadvantages of the method.

**Detailed Comments:**

See the weaknesses section.

**Justification Of The Final Rating:**

Thanks for your response. The response has addressed parts of my concerns and makes the manuscript's clarifications and experiments more comprehensive and appears more reasonable. However, after considering the rebuttal and comments of other reviewer(s), especially from the Reviewer v7SV, I am afraid I have to maintain the preliminary rating.

**Justification Of The Preliminary Rating:**

The paper addresses an important topic and presents ideas towards it. The rating preliminary recognizes the contributions of the approach but also highlights several areas that require further clarification. Addressing these key issues is necessary for strengthening the overall work.

**Questions To Address In The Rebuttal:**

See the weaknesses section.

**Special Issue:**

No

---

> ### Author Response · Authors · 2025-03-08
>
> Thank you for your valuable feedback. We address your concerns as follows:
>
> ### Grad-CAM visualizations
> We have included additional Grad-CAM visualizations in Appendix D, comparing our full model (based on DinoBloom-S) with the baseline DinoBloom-S classifier across different neutrophil abnormalities. We observe that our model (middle column) consistently focuses on morphologically relevant regions, while the baseline model (right column) often attends to irrelevant background areas or fails to concentrate on diagnostically important features.
>
> ### Inference speed and computational resources
> Thank you for your comment. Our method adds minimal extra parameters (~15K for constraint matrix generation, ~21 for thresholds) compared to the DinoBloom-S backbone (22M). For inference speed, MC dropout sampling adds a 2× overhead. Training our full model takes ~30 minutes on a single NVIDIA A100 GPU. During inference, our method requires approximately 0.16 seconds per image when using 50 Monte Carlo samples. This is roughly twice the 0.08 seconds needed for a single forward pass, reflecting the additional stochastic evaluations required to estimate predictive uncertainty. However, we find this overhead acceptable given the improved performance and the valuable confidence measures obtained. A detailed ablation for the influence of MC is presented in Appendix, Table 9.
>
> | MC Samples | Accuracy (%) | Macro-F1 (%) | Mean Unc. (%)| Inference Time x |
> |------|--------------|--------------|--------------|--------------|
> | 1 (no MC)    | 88.2         | 92.4         | --   | 1.0x |
> | 10    | 91.5        | 90.5         | 3.1      | 1.3x |
> | 20   | 92.0         | 91.0         | 2.8      | 1.6x |
> | 50    | 93.0         | 92.0         | 2.4     | 2.0x |
> | 100    | 93.3         | 92.2         | 2.3   | 4.0x |
>
> Alongside the time cost, we also examined the approximate FLOPs needed for inference. Using 50 Monte Carlo samples effectively doubles the FLOPs compared to a single forward pass of the chosen backbone, consistent with the increased number of forward evaluations. In our typical implementation, our method runs at about $2 \times 10^{11}$ FLOPs per image with 50 samples (vs. $1 \times 10^{11}$ FLOPs for a single pass), though exact figures may vary depending on hardware optimizations and specific backbone architectures. As before, reducing the Monte Carlo samples proportionally lowers the FLOPs, allowing a trade-off between computational overhead and uncertainty quantification. We added Appendix F to include this additional discussion about the complexity of our method.
>
> ### Hyperparameter sensitivity
> We've added an ablation study on the λ-hyperparameters in Appendix E.3 showing that small variations to these parameters result in only minor performance changes (within 1-2% of baseline). More importantly, our ablation reveals the differential impact of each component: removing the constraint loss ($\lambda_1=0$) causes the most significant drop (8%), removing uncertainty awareness ($\lambda_2=0$) reduces performance by 4.3%, while removing entropy regularization ($\lambda_3=0$) has the smallest impact (1.5% reduction). We've also tested with 5 different random seeds, finding consistent results (std. dev. of ±0.6% in accuracy). Please refer to the Appendix and comment 4 of the reviewer UCa8 for additional results.
>
> ### Additional backbones
> For our experiments, we focused on DinoBloom since this model is trained for single-cell images in hematology, and it is the most appropriate model for our application. However, as you mentioned, our constraint module is architecture-agnostic and can work with any feature extractor. In this revised version, we have included the performance using all the foundational models, of the the original table (Table 1), as backbones, observing similar relative improvements. The constraint module is architecture-agnostic and can work with any feature extractor.

---

### Official Review · Reviewer_UCa8 · 2025-02-28

**Confidence:** 4
**Preliminary Rating:** 4
**Recommendation:** Oral
**Final Rating:** 5

**Summary:**

This paper proposes a modular approach to enforce biological constraints in multi-label medical image classification, addressing the issue of biologically impossible predictions. The method integrates learnable constraint matrices and adaptive thresholding to prevent contradictory predictions. Experiments on three datasets (two public, one in-house) demonstrate improvements over existing foundation models and state-of-the-art methods. Detailed ablation studies and additional analyses suggest that the approach successfully captures biological relationships between abnormalities.

**Strengths:**

The method seems sound. The proposed approach (learnable constraint matrices + adaptive thresholding) is well-motivated and logically structured.
Strong empirical results are obtained. The method significantly outperforms state-of-the-art foundation models across multiple datasets.
The method explicitly incorporates domain knowledge into the classification process, addressing a real clinical need.
The paper includes in-depth analyses demonstrating the effectiveness of each proposed component through ablation studies and interpretability.
Uncertainty estimation and threshold behavior analyses add robustness to the findings.

**Weaknesses:**

Motivation for constraints: While the paper explains the need for enforcing biological constraints, it does not explain why self-supervised learning (SSL) alone fails to capture these relationships.
"they do not explicitly incorporate domain-specific constraints or handle biologically impossible co-occurrences." Develop on this. Many things are not explicitly incorporated in SSL, yet they are learned. It actually makes the strength of SSL, implicitly learning some structure of the data. Explain why this is more problematic in your scenario.
Do we not have enough data to learn this and so require prior info ?
Or is it too complicated to learn it and so deteriorates training ?
Models trained on natural images learn the probabilities of class co-occurrence and class similarities without explicit constraints.

Incomplete related work: The idea of modeling label co-occurrence is not unique to medical imaging. The authors should present other works in CV and ML and compare their method to similar approaches (not necessarily obtaining results on the datasets, but commenting).

In-house dataset patient information: It is unclear why patient-level tracking is not possible. "without any patient level information." How is that possible ? You cannot split by patients ? And don't even know how many patients it originates from ?

Hyperparameter robustness: The model has a relatively large number of hyperparameters, but their selection and the robustness to variations are not sufficiently discussed. The abstract mentions "hyperparameter interpretation", I am not sure what it refers to.

MC dropout inference speed: The impact of Monte Carlo dropout on inference speed should be discussed.

**Detailed Comments:**

Throughout the intro, the motivation for constraints (in general) are well stated. However, while the "learnable constraint matrices and adaptive thresholding" are repeated multiple times, it is not clear what these methods achieve and why they are needed. A brief motivation/explanation of those would be useful in the intro instead of repeating.
The paper should explain why SSL alone does not learn these constraints effectively. Is it due to insufficient data or the complexity of relationships?
Related work should include CV/ML methods that model label dependencies and highlight why existing methods are insufficient.

Class imbalance handling:  It reads as if you over-sample minority and under-sample majority, then still use class-weights as if it wasn't over/under-sampled. Just clarify.

Table clarity issues:
	Table 1: incorrect 2nd best are underlined. Also best for Hypogr. This is misleading, even though the results are very good anyway.
	Tables 1 and 2: the backbone of your method is ViT-S right? mention it for better comparison with the corresponding method.

Interpretability of attention maps: It is not obvious to me that the attention maps show that the the model attends to specific morphological features characteristic of each abnormality type.

Citation formatting: The references might not follow the correct MIDL format and need verification.

**Justification Of The Final Rating:**

Even though the authors did not reply to my last question (change of results in Table 1 after revision), I think they significantly revised the manuscript and clarified all other concerns. I suggest acceptance of this very interesting paper.

**Justification Of The Preliminary Rating:**

The problem is clinically relevant. The methodology is sound and results are excelent.
The proposed approach is novel in medical imaging, but modeling label dependencies is not unique to this domain. A stronger connection to existing ML research is needed.
Some clarifications are required.

**Questions To Address In The Rebuttal:**

Include related work outside of medical imaging if relevant.
Why do existing SSL models fail to learn biologically impossible co-occurrences?
Why is patient-level information unavailable in the in-house dataset? How does this affect dataset splitting and generalization?
How sensitive is the model to hyperparameter variations?

---

> ### Author Response · Authors · 2025-03-08
>
> ### Motivation for constraints
>  We agree with the reviewer that SSL is very powerful and the best way to learn some structure of the data implicitly. Using the foundation model DinoBloom, we aim to take advantage of these data-driven self-supervised characteristics that were captured while training on huge databases. However, such training is class-agnostic, and it is mainly focused on the imaging characteristics of the different cells. Our main contribution to this paper is the integration of some additional domain knowledge that incorporates some prior known constraints into the SSL models. In the Introduction, we added discussions emphasizing that self-supervised pretraining, while powerful, is often data-driven and does not encode explicit domain knowledge. In contexts where contradictory abnormalities are rare or subtle, purely SSL can fail to model negative co-occurrences. We reference this explicitly in Section 1, paragraph 1.
>
> ### Incomplete related work regarding label co-occurrence outside medical imaging.
> We updated Related Work with additional references detailing generic multi-label approaches that model label dependencies (e.g., graph-based label co-occurrence, adjacency learning). In particular, we cite additional references (Pham et al., Bruton et al, Wang et al..) and clarify how our method differs by incorporating explicit domain constraints. Due to space constraints we did not add additional references, however in case you think that there are more citations missing we will be happy to incorporate them.
>
> ### In-house dataset patient-level info
> Thank you for your comment. We added clarifications in Section 4.1 (Datasets) stating that the GR-Neutro dataset was de-identified with no patient IDs, preventing a strict patient-level split. Indeed, this study has focused on the accurate classification of different neutrophil morphology analyses and not on patient-level diagnosis. For the construction of the dataset, images extracted from 30 patients were used, and the annotations were performed on high-resolution microscopy images of neutrophils collected by a Sysmex DI-60 system. We therefore used stratified sampling to maintain class distributions on the neutrophil level. However, after your comment, we would like to highlight that patient-level diagnosis is our future work, and we are working on enhancing our dataset with patient-level characteristics.
>
> ### Hyperparameter robustness
> Hyperparameter sensitivity analysis was added in Appendix E using the GR-Neutro dataset. Five random seeds showed minimal variation (±0.6% accuracy, ±0.5% Macro-F1). Adjusting λ-values by ±20% caused minor changes, while individual ablations showed constraint loss is most crucial: removing it (λ₁=0) dropped performance to 85.0%, removing uncertainty awareness (λ₂=0) to 88.7%, and removing entropy regularization (λ₃=0) to 91.5%.
>
> **(A) Random Seeds (5 runs)**
>
> | Seed | Accuracy (%) | Macro-F1 (%) |
> |------|--------------|--------------|
> | 1    | 93.0         | 92.4         |
> | 2    | 94.0         | 93.2         |
> | 3    | 92.8         | 92.0         |
> | 4    | 93.2         | 92.5         |
> | 5    | 93.7         | 92.8         |
> | Mean | 93.3         | 92.6         |
> | Std  | ±0.6         | ±0.5         |
>
> **(B) Varying Hyperparameters**
>
> | $\lambda_1$ | $\lambda_2$ | $\lambda_3$ | Macro-F1 (%) |
> |-------------|-------------|-------------|--------------|
> | 0.1         | 0.1         | 0.01        | 93.0         |
> | 0.12        | 0.12        | 0.012       | 92.3         |
> | 0.08        | 0.08        | 0.008       | 92.0         |
> | 0           | 0.1         | 0.01        | 85.0         |
> | 0.1         | 0           | 0.01        | 88.7         |
> | 0.1         | 0.1         | 0           | 91.5         |
>
> ### MC dropout inference speed
> Thank you for pointing out this. Indeed adding MC dropout with 50 samples increases inference time by approximately a factor of 2 compared to a single forward pass. Specifically, if a single forward pass requires 0.08 seconds for one image, the MC dropout approach with 50 draws requires around 0.16 seconds. Given that MC dropout provides a notable boost in performance (approximately +2% in F1 score) and yields valuable uncertainty estimates (as shown in Figure 3, right), we find the overhead to be acceptable in most clinical contexts. Nevertheless, it is worth emphasizing that our primary contribution is independent of the MC dropout mechanism. To integrate your comment, we included in the implementation details the time for a single forward pass for the MC dropout.

---

> > ### Comment · Reviewer_UCa8 · 2025-03-11
> >
> > Thank you for your detailed response. You clarified all my points and the paper is now clearer, more complete, and the benefits of the approach more striking with the results in table 1.
> > Last question: why did the results for Hyposeg and hypogr change in table 1 ?

---

> > > ### Author Response · Authors · 2025-03-15
> > >
> > > Thank you very much for your comments and review that considerably improved our manuscript.
> > >
> > > **Change on Table 1:** We are really sorry for these typos. We have now corrected the typos on Table 1 to the correct values.

---

> ### Author Response · Authors · 2025-03-08
> **Minor updates**
>
> - **Class imbalance**: To address class imbalance in our dataset, we employed a two-step approach. First, we applied SMOTE to increase the representation of underrepresented classes and random undersampling to reduce the number of majority class samples, to balance class distribution. However, since perfect balance is difficult to achieve through sampling alone (especially for very rare classes), we implemented a second layer of correction through class-specific weights in our loss function. These weights were calculated based on the inverse frequency of each class in the post-sampling training data, giving higher importance to classes that remained underrepresented.
>
> - **Tables**: Corrected underlinings for second-best results; clarified DinoBloom-S backbone
> - **Attention maps**: Added class-specific attention maps in appendix D showing our method focuses on relevant regions compare to the baseline.
> - **Citations**: Formatted per MIDL guidelines

---

### Author Rebuttal · Authors · 2025-03-08

**Rebuttal:**

# Response to Reviewers

We thank the reviewers for their constructive feedback. We provided point-by-point responses to each reviewer's comment and referenced the modifications we have made in the manuscript. All manuscript changes are indicated in **red** in the revised PDF to clarify where we addressed each point. Section headings (e.g., Section 2.1, Appendix A, Table 3) in our responses correspond to the updated version of the manuscript. We believe these revisions have strengthened the paper's contributions, improved its clarity and reproducibility, and addressed all reviewer concerns.

**Supporting Material:**

/attachment/3c54d00e85574231edd4ef654b6494b9dc911870.pdf

---

> ### Author Response · Authors · 2025-03-15
> **Re-upload a revised manuscript during the discussion period**
>
> Dear PCs and ACs,
>
> Is there any option to re-submit our revised paper to address the feedback of the reviewers during the discussion phase? We cannot find this option in the platform.
>
> Thank you very much for your help and assistance.
>
> Best regards,
> The authors

---

### Comment · Program_Chairs · 2025-03-18
**Concerns from authors**

Dear AC,

We received an email raising a concern that one reviewer may fail to check the latest response from the authors. I am copying the email below:

 We just saw that one of our reviewers (the one with the most negative review) updated his/ her score but on the justification he/ she mentioned that we have not addressed the comments during the discussion period. Could you please confirm that our answers are in the openreview platform? We are a bit puzzled with this justification since we put a lot of effort on addressing the comments during the discussion period. Is there any problem from our side? Will our answers be visible to the AC that handle our paper?

Could you please check with the reviewer to confirm that their rating reflects the latest available information? Whether or not the reviewer is responsive, the Area Chair however, are encouraged to still make a decision based not only on the reviewer's rating but also on the overall context.

---

> ### Comment · Area_Chair_HhY4 · 2025-03-18
>
> Hi,
>
> Thanks for checking this.
>
> I think the reviewer has not replied because the response is very close to the discussion deadline. Based on what I can see, the information provided by the authors is fully visible to me. I will also inquire with the reviewer to confirm whether their assessment is final or not.
>
> Best,
> AC

---

### Meta-Review · Area_Chair_HhY4 · 2025-03-21

**Recommendation:** Reject
**Confidence:** 4

**Metareview:**

This paper presents a training strategy for biologically-constrained multi-label classification. The contributions primarily focus on two perspectives: adding regularization terms to the features based on a pre-defined correlation matrix and adaptive thresholding. These two components do not have a direct connection but intuitively fit the task of multi-label classification well. The authors demonstrate the effectiveness of the proposed methods through detailed experiments, including an ablation study.

This manuscript received mixed reviews. Most reviewers appreciated the idea of improving the performance of multi-label classification. However, one reviewer, who was particularly responsible and careful in evaluating this paper, raised several critical points regarding its technical issues.

After carefully reading the reviews, the authors' response, and examining the equations and code myself, I believe this paper is not yet ready for publication. I agree with the reviewer on the following reasons:

1. The descriptions in the method section are inconsistent with the code, particularly in the calculation of R.
2. The choice of using a transformer layer instead of a simple MLP is not convincingly justified.

In addition, I have some concerns after carefully reading the paper:

1. It is unclear why the authors emphasize foundation models. The proposed training strategies are applicable to all training paradigms for multi-label classification and are not specifically tied to foundation models.
2. The meaning of the uncertainty U is not entirely clear. It appears to have different interpretations and statistical properties when used during training versus inference. The authors state that they calculate uncertainty in both training and inference, which makes its role vague. For example, if it is calculated during inference, it cannot be optimized during training. If they refer to different concepts, they should be denoted with different symbols.

In summary, although I agree with most reviewers that this paper has merits, particularly due to the novelty of incorporating biological constraints, the paper cannot be accepted in its current form. The vague and inconsistent descriptions are likely to confuse careful readers and potentially mislead the community. I encourage the authors to refine their work further, both in terms of model design and methodological clarity, before submitting it to another venue.